# Comparison of Yeast and CHO Cell-Derived Hepatitis B Vaccines and Influencing Factors in Vaccine-Naïve Adults in China: Insights for Personalized Immunization Strategies

**DOI:** 10.3390/vaccines13030295

**Published:** 2025-03-10

**Authors:** Qian Qiu, Huai Wang, Wei Zhang

**Affiliations:** Beijing Center for Disease Prevention and Control, Beijing 100013, China; qiuq@bjcdc.org (Q.Q.); wanghuai@bjcdc.org (H.W.)

**Keywords:** hepatitis B virus, immunization, adults, influencing factors

## Abstract

Background: Various factors influence the immunologic responses to HBV vaccines in adults, including unchangeable individual characteristics. Personalized vaccination regimens accounting for host factors can enhance immune efficiency, particularly for adults at higher risk. Methods: In this two-center controlled trial, HBV vaccine-naïve participants aged 25–55 were randomly administered the two types of HBV vaccines (yeast cell-derived (YDV) or Chinese hamster ovary (CHO) cell-derived) at 0–1–6 months. Antibody titers were measured eight weeks after the final dose. Results: Overall, 289 participants with YDV and 293 participants with CHO completed the three-dose series and antibody testing. The seroprotection rates (SPRs) were comparable (97.23% vs. 98.98%; *p* = 0.1398), but the geometric mean concentration (GMC) was significantly higher for the CHO (1627.83 mIU/mL vs. 600.76 mIU/mL; *p* < 0.0001). The GMC of both regimens declined significantly in individuals aged ≥45 years and males. Unlike the YDV, the GMC of CHO was minimally affected by BMI or smoking or drinking status. Conclusion: The CHO regimen may be advantageous for HBV vaccine-naïve adults aged 25–55 with BMI ≥ 25 or those who smoke or drink, in terms of immunogenicity and durability, providing insights for personalized immunization strategies.

## 1. Introduction

Infection with hepatitis B virus (HBV) is a major public health concern. It was estimated that approximately 254 million people were chronically infected in 2022, with 1.2 million new infections each year. Hepatitis B resulted in almost 1.1 million fatalities due to HBV-related sequelae, including cirrhosis, chronic hepatitis, and cancer in 2022 [1]. Since the 1990s, the World Health Organization (WHO) has advocated for hepatitis B vaccination as a principal technique for preventing HBV infection, and the vaccine’s efficacy in adults has been well-documented in previous studies.

Vaccine-related factors, such as antigen type, dosage, immunization schedule, and the adjuvants used, have been proven to affect immunological responses to HBV vaccines in adults, as demonstrated in existing studies [2,3]. In China, two types of recombinant HBV vaccines are primarily used: those derived from yeast cells (Hansenula polymorpha and Saccharomyces cerevisiae, YDV) and those derived from mammalian cells (Chinese hamster ovary cells, CHO). For the YDV, the S sequence, subtype (adw2), is inserted into the expression vector pMPT121 and transformed into the *H. polymorpha* host strain RB10, offering potential advantages due to the ability to perform complex glycosylation, the availability of genetic tools, thermotolerance, and the utilization of various carbon sources [4]. Meanwhile, for the CHO, a recombinant plasmid pSHBVB813 containing a PvuII fragment of HBV-adw DNA in plasmid pSP65 and the pSVE100-DHFR expression vector is used [5]. Both vaccines use the same adjuvants, Al(OH)_3_ and NaCl. The CHO-derived vaccine generates a recombinant HBsAg structurally resembling that of the wild virus, with a structure more similar to native HBsAg than that of the yeast-derived vaccine, including all three of the envelope proteins (weight of the total protein: S peptides, 76.3%; pre-S2, 21.7%; and pre-S1, 2%), whereas the yeast-derived vaccine consists solely of the S protein [5]. As a result, the CHO-derived vaccine elicits a quick and strong antibody response, likely due to its molecular weight, genetic sequence, size, and the quantity of HBsAg monomers [6]. Therefore, vaccinations including pre-S/S proteins may enhance immunization efficacy in patients who have previously displayed inadequate or absent responses to conventional immunizations [7].

In addition to vaccine-related factors, which can be adjusted to enhance immune efficiency, numerous studies have shown that the immune response induced by the vaccine may be influenced by the host factors [3,8], including age, sex, smoking or drinking status, and overweight or obesity, which cannot be modified through vaccination. Therefore, incorporating host factors into personalized vaccination regimens—particularly for individuals who are more likely to have a poor or no response—may enhance immune responses and augment vaccination effectiveness.

Here, the immunogenicity of two types of recombinant HBV vaccines commonly used in China was evaluated among vaccine-naïve adult individuals between the ages of 25 and 55. The study sought to explore alternative, individualized vaccination approaches for adults with different statuses in China to achieve adequate immune responses and long-term protection against HBV.

## 2. Materials and Methods

### 2.1. Study Design

This prospective, randomized, two-center clinical investigation was carried out at two locations in China—Huaibei County in Anhui Province and Xuanhua County in Hebei Province—from 2013 to 2014. Prior to participation, participants were tested for HBV surface antigens (HBsAg) and antibodies recognizing the HBV core antigen (anti-HBc) and HBsAg (anti-HBs). Participants testing negative for all these markers were randomly assigned to one of two groups through the use of a computer-generated code, ensuring an unbiased allocation process. Participants also completed a questionnaire that gathered information on age, sex, body mass index (BMI), personal health data (including smoking and alcohol consumption history), and hepatitis B vaccination history. Body BMI was divided into three categories based on WHO guidelines [9]: underweight/normal weight (≤24.9), overweight (25.0–29.9), and obesity (≥30). Smoking or alcohol consumption history was defined as current or previous consumption of cigarettes or alcohol. Blood samples for immunological assessments were obtained eight weeks after the administration of the final vaccination, allowing for sufficient time to assess immune responses. Figure 1 represents the study design and flow.

All participants provided written informed consent. The study protocol was approved by the medical ethics committees of Huaibei People’s Hospital and The Second Affiliated Hospital of Hebei North University. The study was conducted in accordance with Good Clinical Practice Guidelines and adhered to the ethical principles outlined in the Helsinki Declaration (clinical trial registration: Identifier No. NCT06515938).

### 2.2. Study Population

The following are the criteria that were used to participate in the study: (a) individuals between the ages of 25 and 55; (b) seronegative for HBsAg, anti-HBs, and anti-HBc; and (c) no previous immunization against HBV or the hepatitis A virus. Included in the list of exclusion criteria were the following: (a) a history of allergies to vaccine components; (b) autoimmune disorders or immunodeficiency; and (c) acute illnesses within the last week, recent immunizations (during the past month), or fever (an axillary temperature > 38 °C) within the previous three days.

### 2.3. Vaccines and Vaccination

The two recombinant vaccines of hepatitis B that were used in the study are commonly available in the Chinese market. The participants were randomly divided into two vaccination groups in a 1:1 ratio: (1) Group I: Kangtai recombinant yeast-derived hepatitis B vaccine (batch no: B201202006; Shenzhen Kangtai Biological Products Co., Ltd., Shenzhen, China); and (2) Group II: Huabei recombinant hepatitis B vaccine produced in CHO cells (batch no: 201303Y1306; North China Pharmaceutical, Jintan Biological Products Co., Ltd., Shijiazhuang, China). Each vaccination contained 20 µg of HBsAg per 1 mL dosage. It was delivered intramuscularly into the upper deltoid muscle at time intervals of 0 months, 1 month (30 days following the initial dose), and 6 months (180 days following the initial dose). Each dose was administered to alternating arms.

### 2.4. Serological Tests

The levels of HBsAg, anti-HBs, and anti-HBc were determined prior to vaccination, while anti-HBs titers were tested eight weeks following the third vaccination using the Architect i2000 system (Chemiluminescence Microparticle Immunoassay, Abbott, Chicago, IL, USA). The range for anti-HBs detection was 1–1000 mIU/mL. The sera were diluted for anti-HBs quantitation.

### 2.5. Statistical Analyses

The sample size was estimated with the two rates comparison formula, *n*_1_ = *n*_2_ = 0.5[$\frac{z_{\alpha }+z_{\beta }}{{sin}^{-1}\sqrt{p_{1}}-{sin}^{-1}\sqrt{p_{2}}}$]^2^ (*α* = 0.05, *β* = 0.1), with the assumption from previous reports that the seroprotection rate (SPR) for the two groups was 99% (*p*_1_) and 95% (*p*_2_), respectively. Each group included an estimated 273 samples.

The primary outcomes were vaccine-induced seroprotection (anti-HBs levels ≥ 10 mIU/mL 8 weeks post-vaccination) [10] and the percentage of individuals who obtained the seroprotection.

The secondary outcomes included the anti-HBs category response eight weeks post-third vaccination: a concentration of 10 to 100 mIU/mL was classified as a low reaction, 100 to 1000 mIU/mL as a moderate response, and ≥1000 mIU/mL as a hyper-response [11]. Anti-HBs titers were subjected to log transformation to determine geometric mean concentrations (GMCs), with 95% confidence intervals (95% CI) provided. Descriptive analysis within groups was conducted using Pearson’s *χ*^2^ test, *χ*^2^ for trend, Fisher’s exact test for categorical variables, and *t*-tests for continuous variables. *p*-values < 0.05 (two-tailed) were deemed statistically significant. All statistical analyses were performed using SPSS 18.0.

## 3. Results

### 3.1. Demographics Analysis

Two vaccine groups were assigned at random to a total of 761 people, with 367 receiving the YDV regimen and 394 receiving the CHO regimen. After completing the series of three vaccinations and serological testing, 289 individuals in the YDV group and 293 in the CHO group had completed the immunization series eight weeks following the last injection. An insignificant difference in demographic characteristics was observed between the two regimens (Table 1).

### 3.2. Immunogenicity

Eight weeks following the final injection, the SPR showed an insignificant difference between the YDV group and the CHO group (97.23% vs. 98.98%; *p* = 0.1398). However, the GMCs differed significantly: the YDV regimen had a lower GMC of 600.76 mIU/mL (95% CI: 495.27–728.65 mIU/mL), while the CHO regimen had a higher GMC of 1627.83 mIU/mL (95% CI: 1332.75–1986.24 mIU/mL) (*p* < 0.0001). The category evaluation of anti-HBs concentrations revealed a marked compositional difference (*p* < 0.0001): in the YDV group, 9.61% had a low response, 49.11% displayed a moderate response, and 41.28% had a hyper-response. In contrast, the CHO group showed 5.17% with a low response, 25.17% with a moderate response, and 69.66% with a hyper-response.

### 3.3. Immune Response by Demographic Factors

#### 3.3.1. Immune Responses in Different Age Subgroups

In the CHO group, participants under 45 years achieved a significantly higher SPR compared to those over 45 years (*p* = 0.0156), while the YDV group did not show a significant age-related difference (*p* = 0.1049). Both the YDV and CHO groups demonstrated an age-related decline in GMCs (*p* = 0.0088 and *p* = 0.0013, respectively) (Table 2).

The two age groups showed comparable SPRs between the YDV and the CHO groups (*p* = 0.0611 and *p* = 0.7151, respectively). GMCs, however, were markedly raised in the CHO cohort for both age groups (*p* < 0.0001 and *p* = 0.0129, respectively) (Table 2). The anti-HBs levels also differed markedly between the two age groups, representing significantly different antibody category responses. The CHO regimen showed a higher proportion of hyper-responses and lower numbers of low responses (<45 years: *p* < 0.0001, ≥45 years: *p* = 0.0293) (Figure 2).

#### 3.3.2. Sex-Based Differences in Immune Response in Adults

No significant sex-related differences in SPR were observed between the two vaccine groups (*p* = 0.3020 and *p* = 0.5791, respectively). However, men showed markedly lower GMCs relative to women in both the YDV and CHO groups (*p* = 0.0057 and *p* = 0.0044, respectively) (Table 2).

Both male and female participants showed comparable SPRs between the two vaccine groups (*p* = 0.4465 and *p* = 0.3667, respectively). In contrast, GMCs were markedly higher in the CHO groups for both sexes (*p* < 0.0001) (Table 2).

Further analysis of antibody category responses revealed significant differences in anti-HBs concentration distribution between males and females in both vaccine cohorts (YDV: *p* = 0.0145; CHO: *p* = 0.01224). The CHO vaccine elicited higher numbers of hyper-responses and a lower proportion of low responses across both sexes (*p* < 0.0001) (Figure 2).

#### 3.3.3. Immune Responses in Relation to BMI

Among participants receiving YDV, both SPR and GMC varied significantly across different BMI subgroups, with overweight or obese individuals showing lower antibody responses compared to those with underweight or normal BMI (*p* = 0.0155 and *p* = 0.0017, respectively). In contrast, insignificant BMI-related differences in SPR or GMC were observed in the CHO group (*p* = 0.7935 and *p* = 0.6019, respectively) (Table 2).

SPRs were insignificantly different between the two vaccine groups across BMI subgroups (underweight or normal: *p* = 0.2408; overweight: *p* = 0.2452; obesity: *p* = 0.1010). However, participants in the CHO regimen achieved significantly higher GMCs in all three BMI subgroups (*p* = 0.0016, *p* = 0.0001, and *p* < 0.0001, respectively). Details are provided in Table 2.

The percentage of category responses in the YDV group differed significantly among participants across the three BMI subgroups (*p* = 0.0028), while an insignificant difference was observed in the CHO cohort (*p* = 0.2081). This cohort also showed a higher proportion of hyper-responses and a lower proportion of low responses across all BMI subgroups (underweight or normal: *p* = 0.0176; overweight: *p* = 0.0003; obesity: *p* < 0.0001) (Figure 2).

#### 3.3.4. Immune Responses of Participants with Smoking History

This study included 154 participants with a history of smoking, with 76 and 78 in the YDV and CHO groups, respectively.

Among participants with a history of smoking, the SPR was similar between the two vaccine groups (*p* = 0.6176). However, the CHO group displayed significantly higher GMCs (*p* < 0.0001) and antibody category responses (*p* < 0.0001) relative to the YDV group (Table 2, Figure 2).

When compared to participants without a smoking history, the groups did not differ significantly in terms of SPR. Nonetheless, among participants with a smoking history in the YDV group, GMCs were significantly decreased (*p* = 0.0084), and the antibody category response showed a less favorable distribution (low response: 17.57% vs. 6.76%; moderate response: 54.05% vs. 47.34%; hyper response: 28.38% vs. 45.89%, *p* = 0.0037). In contrast, the CHO group demonstrated no significant differences in GMCs (*p* = 0.1161) or antibody category responses (low response: 9.09% vs. 3.76%; moderate response: 25.97% vs. 24.88%; hyper response: 64.94% vs. 71.36%, *p* = 0.1760) between participants with and without a smoking history (Table 2, Figure 2).

#### 3.3.5. Immune Response of Participants with Drinking History

This study included 154 participants with a history of alcohol consumption, with 80 in the YDV group and 74 in the CHO group. Among participants with a drinking history, the SPR was insignificant between the two vaccine groups (*p* = 0.3687). However, the CHO group demonstrated significantly higher GMCs (*p* < 0.0001) and antibody category responses (*p* < 0.0001) vs. the YDV cohort (Table 2, Figure 2).

There were no significant variations in SPRs found between the individuals in either group who had a history of alcoholism and those who did not have such a history. In the YDV group, participants with a drinking history showed significantly lower GMCs (*p* = 0.0268) than those without, although the antibody category response was comparable (low response: 13.16% vs. 8.29%; moderate response: 53.95% vs. 47.32%; hyper response: 32.89% vs. 44.39%, *p* = 0.1623). In the CHO group, insignificant differences in GMCs (*p* = 0.9493) or antibody category responses (low response: 5.48% vs. 5.07%; moderate response: 27.40% vs. 24.42%; hyper response: 67.12% vs. 70.51%, *p* = 0.8610) were observed between participants with and without a drinking history (Table 2, Figure 2).

## 4. Discussion

The objective of the current research was the assessment of the immunogenicity of two hepatitis B vaccines, each containing 20 μg, which were produced from yeast and CHO cells, in vaccine-naïve individuals between the ages of 25 and 55. Given that vaccine-induced protection can vary depending on individual factors, the goal was to explore personalized vaccination strategies to enhance efficacy and prolong protection. The results demonstrated that both vaccines elicited effective protective immunity, with an SPR of 97.23% for the YDV regimen and 98.98% for the CHO regimen. The CHO regimen resulted in significantly higher anti-HBs titers at eight weeks after the third dose, with GMCs of 1627.83 mIU/mL, relative to 600.76 mIU/mL for the YDV regimen.

The two vaccines used in this study were highly effective in previous research; however, the results have been inconsistent. Wu et al. performed a systematic review and meta-analysis evaluating immunization programs in China, which found no marked differences in SPR between CHO-derived and yeast-derived vaccines (Risk Ratio: 1.01, 95% CI: 0.98–1.04) [12]. It was concluded that both types of vaccines were suitable for adults in China. In contrast, research by Vesikari et al. identified that the hepatitis B vaccine containing the S, pre-S1, and pre-S2 antigens showed greater efficacy than the vaccine containing the single S antigen. The SPR in subjects aged 18 and older was 91.4% for the former and 76.5% for the latter, measured four weeks after the third vaccination [7]. The discrepancies in response across studies might be due to differences in participant demographics, such as age, smoking and drinking habits, obesity, sex, and concurrent diseases, all of which can affect vaccine-induced immunogenicity [3]. Furthermore, variations in study design and the types of vaccines used may also contribute to the differing outcomes.

Previous studies indicate that HBV transmission can occur when anti-HBs titers are inadequate. Stramer et al. reported that a low vaccine-induced anti-HBs titer (<100 mIU/mL) might be insufficient to avert HBV infection over an extended period with non-A2 HBV subgenotypes [13], including genotypes B and C, which were most common in Asia [14]. As anti-HBs titers induced by vaccines decline gradually with time, occasionally even becoming undetectable, achieving a higher antibody titer following vaccination is crucial to maintaining protective levels and ensuring durable protection against potential HBV infection [15,16]. Therefore, vaccines capable of generating and sustaining higher antibody titers are desirable for prolonged protection. In the present study, while the SPR was similar between the two vaccine groups, the CHO vaccine induced a higher proportion of hyper-responses (>1000 mIU/mL) in 69.66% of recipients and a lower proportion of low responses (10–100 mIU/mL) in only 5.17% of recipients. In contrast, the YDV group showed 41.28% hyper-responses and 9.61% low responses. These findings suggest that the CHO vaccine offers greater potential for maintaining antibody levels over time, making it an appealing option for long-term protection against HBV.

The present study demonstrated that young adults (aged <45 years) display stronger antibody responses to HBV vaccines compared to older individuals (aged ≥45 years). Similarly, another study aligned with these findings, reporting better antibody responses in younger individuals [2,3,17]. The lower responsiveness observed in older adults may partly result from a decline in cellular responses and weakened humoral immune function associated with advancing age [18,19]. Furthermore, the present study found that the male sex was linked to a lower antibody response, consistent with findings from several previous studies [20,21]. This difference may result from the specific actions of sex hormones on genetic regulation. Estrogen can stimulate monocytes to release IL-10, which promotes B lymphocytes to secrete IgM and IgG [22]. In contrast, testosterone suppresses IgG and IgM secretion by B cells and inhibits IL-6 secretion by monocytes [23]. Furthermore, differences in gene structure between females and males may partially explain the observed sex-related heterogeneity in vaccine response. Females have a higher number of immunological genes on the X chromosome compared to males on the Y chromosome [24]. Furthermore, females show stronger immune memory, enabling a rapid induction of high levels of anti-HBs following vaccination [25]. Based on our findings, future strategies should prioritize older adults and males, as these groups are more susceptible to the virus and disease progression. They also tend to show suboptimal responses to both the YDV and CHO vaccines, resulting in reduced duration of protection and a higher risk of potential infection.

A significant body of published data suggests that overweight or obesity is associated with reduced antibody responses compared to non-obese individuals [26,27,28]. Present findings from the YDV vaccine group align with these observations. This reduced responsiveness may be attributed to inflammation induced by leptin in B cells [29], as well as compromised T-cell activities and lymphocyte proliferation [30]. An additional explanation is that vaccines may primarily distribute into fat tissue rather than muscle, potentially hindering absorption and leading to antigen denaturation due to enzymatic activity [31]. Vitamin D deficiency could also contribute to the impaired vaccine response observed in overweight or obese individuals. Vitamin D is essential for promoting the maturation and function of key immunological cells and for reducing inflammation [32]. Furthermore, we observed that individuals with a history of smoking in the YDV group displayed lower GMCs of anti-HBs antibodies. This aligns with a 20-year cohort investigation in which it was found that smoking has an adverse influence on antibody responses, both early and long-term, to hepatitis B vaccines given in infancy [33]. Some immune cells, such as B cells, T helper cells, memory T and B lymphocytes, macrophages, dendritic cells, and natural killer cells, have been shown to be impacted by smoking, which affects both innate and adaptive immunity [34]. It also disrupts cellular and humoral immune responses, with nicotine impairing antigen-mediated pathways and the intracellular calcium signaling required for T-cell activation, thereby inhibiting the antibody response [8]. Similarly, studies have suggested that alcohol consumption may suppress vaccine-induced immune responses [35]. Like smoking, alcohol significantly suppresses the activity of multiple immune cells [36], and its inhibition of B-cell function may contribute to a reduced production of HBV antibodies [37]. The decrease in B cell counts caused by alcohol may hinder antigen presentation, hence weakening immune responses, as B cells are also involved in antigen presentation [38]. Our data indicated a significant difference in vaccination response between those who drank alcohol/smoked and non-consumers within the YDV group. For the CHO group, however, due to the advantage of the preS domains containing several Th cell epitopes, whereas the small HBs protein is a weak Th antigen, the immunosuppressive effect of smoking or alcohol consumption could be partially counteracted. Further studies need to be conducted to quantify the exact relationship with the immune responses as the frequency or intensity of smoking or drinking habit was not captured in this study. Our study revealed that factors such as age ≥ 45 years and male sex significantly reduced the GMC for both the YDV and CHO vaccines. However, antibody levels induced by the CHO vaccine were less affected by factors such as BMI, smoking, or alcohol consumption, suggesting that the CHO vaccine may have broader applicability across different demographic groups. Present evidence suggests that the three-dose 20 μg CHO regimen is advisable for overweight people (BMI ≥ 25), those who smoke, and those who drink due to its superior antibody response and longer immune duration. However, the YDV regimen may still be worth considering for its optimal response, despite the lower antibody titers it induces.

The study has some limitations: (1) Blood samples were not collected after each vaccine dose to monitor detailed trends in immune response. (2) Constrained by the length and funding of the project, blood samples of additional time points (e.g., one month after each dose or long-term follow-up) were not collected to assess antibody persistence; future longitudinal studies need to be conducted to confirm which regimen confers more durable immunity for long-term protection. (3) The immunization history was obtained using self-reported questionnaires, potentially introducing memory bias. (4) Specific amounts of smoking and alcohol consumption were not recorded in this study, limiting the ability to quantify their exact relationship with the antibody response. (5) We excluded participants with major comorbidities considering that they usually had complicated or inconstant status due to diverse pathogenesis; however, other low-level comorbidities or lifestyle factors (e.g., chronic conditions that did not exclude participation, dietary variations) could still be potential confounders for immune response in this study.

## 5. Conclusions

In conclusion, both the three-dose 20 μg CHO regimen and the YDV regimen achieved the desired SPR, exceeding 95%, in HBV vaccine-naïve individuals between the ages of 25 and 55. The CHO regimen resulted in a higher GMC compared to the YDV regimen, suggesting a longer duration of protection. These findings highlight the CHO regimen as the preferred option for susceptible populations with BMI ≥ 25 and those who smoke or drink due to its stronger induction of immune responses and long-lasting immunity. However, for individuals over 45 years of age or males, who tend to have a reduced antibody response to HBV vaccination, additional strategies to strengthen immunity should be prioritized for improved response.

## Figures and Tables

**Figure 1 vaccines-13-00295-f001:**
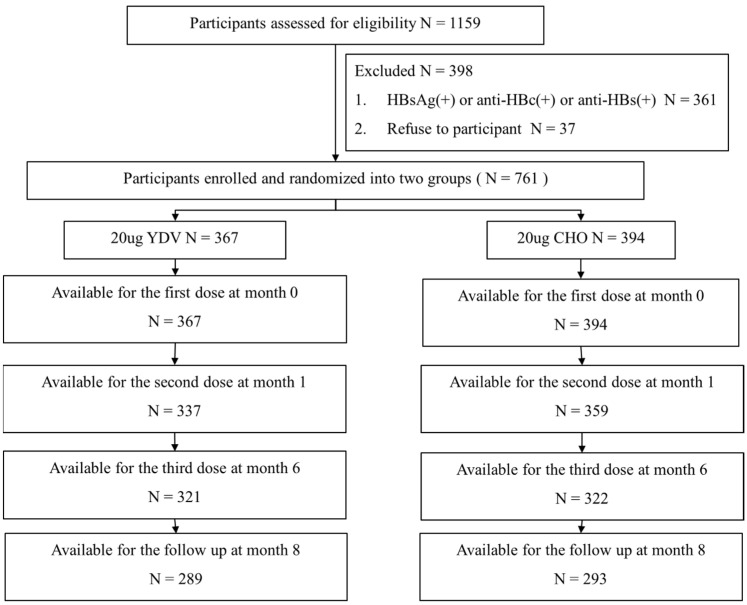
Flowchart of the participant selection process.

**Figure 2 vaccines-13-00295-f002:**
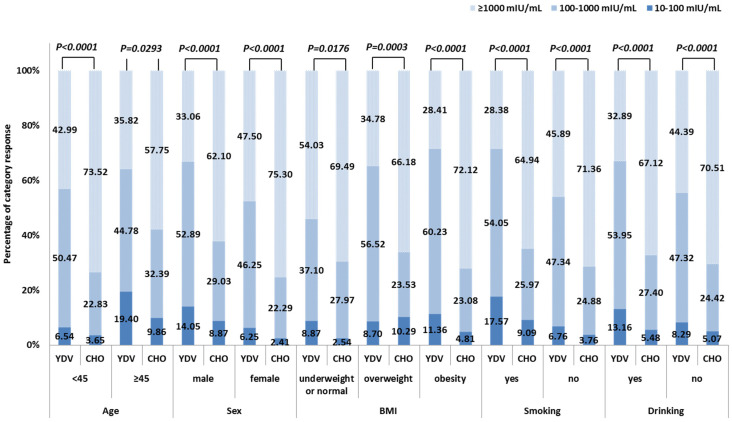
Percentage of responses in each category for the YDV and CHO hepatitis B vaccination regimens, based on participant characteristics.

**Table 1 vaccines-13-00295-t001:** Demographics of participants.

	YDV *	CHO ^#^	*p*-Value
Subjects, N	289	293	
Age, mean (SD)	40.3 (7.1)	39.7 (7.3)	0.3598
<45 years, N (%)	218 (75.4)	219 (74.7)	0.8477
≥45 years, N (%)	71 (24.6)	74 (25.3)	
Sex, N (%)			
Male	126 (43.6)	125 (43.0)	0.8848
Female	163 (56.4)	167 (57.0)	
BMI, mean (SD)	23.8 (3.1)	24.0 (2.9)	0.4236
History of smoking, N (%)	76 (26.3)	78 (26.6)	0.9295
History of drinking, N (%)	80 (27.7)	74 (25.3)	0.5072

* YDV: yeast cell-derived HBV vaccine. ^#^ CHO: Chinese hamster ovary cell-derived HBV vaccine.

**Table 2 vaccines-13-00295-t002:** Analysis of SPRs and GMCs by demographic features for YDV and CHO regimens.

	SPR ^@^ (%)			GMC ^%^ (mIU/mL) (95%CI)		
	YDV *	CHO ^#^	*p*-Value	YDV	CHO	*p*-Value
Age						
<45	98.17	100.00	0.0611	695.06 (567.93–851.50)	1962.55 (1610.02–2397.06)	<0.0001
≥45	94.37	95.95	0.7151	383.75 (238.89–615.85)	931.69 (555.57–1562.43)	0.0129
*p*-Value	0.1049	0.0156		0.0088	0.0013	
Sex						
Male	96.03	98.41	0.4465	441.86 (321.50–606.68)	1167.90 (833.81–1637.62)	<0.0001
Female	98.16	99.40	0.3667	762.04 (602.45–963.91)	2090.17 (1649.13–2649.17)	<0.0001
*p*-Value	0.3020	0.5791		0.0057	0.0044	
BMI ^&^						
Underweight or normal	100.00	98.33	0.2408	880.95 (671.83–1155.17)	1574.98 (1147.11–2162.46)	0.0061
Overweight	95.83	100.00	0.2452	525.84 (356.74–776.66)	1601.99 (1013.33–2532.60)	<0.0001
Obesity	94.62	99.05	0.1010	399.81 (278.11–574.21)	1704.45 (1243.89–2337.88)	<0.0001
*p*-Value	0.0155	0.7935		0.0017	0.6019	
Smoking						
Yes	97.37	98.72	0.6176	390.33 (262.18–580.56)	1248.88 (845.56–1842.72)	<0.0001
No	97.18	99.07	0.1741	700.64 (563.41–872.18)	1791.84 (1418.00–2261.99)	<0.0001
*p*-Value	0.9326	0.7915		0.0084	0.1161	
Drinking						
Yes	95.00	98.65	0.3687	423.27 (281.74–635.24)	1608.41 (1061.04–2440.60)	<0.0001
No	98.09	99.09	0.4399	686.77 (553.36–853.21)	1632.72 (1298.55–2054.94)	<0.0001
*p*-Value	0.2232	0.7462		0.0268	0.9493	

^@^ SPR: seroprotection rate. ^%^ GMC: geometric mean concentration. * YDV: yeast cell-derived HBV vaccine. ^#^ CHO: Chinese hamster ovary cell-derived HBV vaccine. ^&^ BMI: only eight and five participants were underweight (BMI < 18.5) in the YDV and CHO, respectively, thus no separate analysis was performed, as it would not be useful due to low numbers.

## Data Availability

The de-identified data are available upon reasonable request from the corresponding author.

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
