# Peer review of "Comparison of Yeast and CHO Cell-Derived Hepatitis B Vaccines and Influencing Factors in Vaccine-Naïve Adults in China: Insights for Personalized Immunization Strategies"

_vaccines, 2025, doi:10.3390/vaccines13030295_

Round 1
Reviewer 1 Report
Comments and Suggestions for Authors
Vaccines-345390 Evaluation report
The article by Qiu, and Zhang, is entitled “ Efficacy Assessment of Hepatitis B Vaccines and Influencing Factors in Vaccine-Naïve Adults Aged 25–55 in China: Insights for Personalized Immunization Strategies”. In order to investigate alternative personalized HBV immunization tactics, this study assessed the immunogenicity of two different HBV vaccine types-yeast cell-derived (YDV) and Chinese hamster ovary (CHO) cell-derived- vaccines in individuals aged 25 to 55 who had never had an HBV vaccination. The study concludes that, in terms of immunogenicity and durability, the CHO regimen may be beneficial for HBV vaccine-naïve persons aged 25–55 with a BMI ≥ 25 or who smoke or drink, offering insights for individualized immunization protocols. The manuscript is well-written and well-organized with minor editorial issues. However, the following concerns must be addressed by the authors before consideration for publication.
Major concerns:
1.
Figure 2 must be modified to include non smokers and non alcoholics as shown for the age, gender and BMI categories. Also, the legend should be more detailed and it is better to use contrasting colors or patterns for better clarity/presentation.
2.
I think, in Line 16, the number 294 should be 293 as shown in figure 1 and the rest of the text.
3.
Language needs some minor edits such as using the present tense in line 11. Also, in line 58, “was” was used instead of “were”. In addition, “achieve” in line 343 must be “achieved”.
Minor comments:
1.
Acronyms e.g., RR. in line 255…etc. should be spelled out and abbreviated at their first appearance and used as abbreviated later.
2.
Line 37: Several references are required instead of citing only one reference. The same applies for line 287; where several studies were mentioned.
3.
Lines 151-152, please add a p value!
4.
Line 266: Please change “antibody titers are inadequate” to “anti-HBs antibody titers are inadequate” or “HBV-specific antibody titers are inadequate”.
5.
Line 311: Please change “displayed lower GMC of antibodies” to “displayed lower GMCs of anti-HBs antibodies”.
Please see comments
Reviewer 2 Report
Comments and Suggestions for Authors
Thank you for the opportunity to review this manuscript. It addresses an important topic: comparing immunogenic responses to yeast‐derived (YDV) versus CHO‐derived (CHO) recombinant hepatitis B vaccines in a relatively large cohort of vaccine‐naïve adults. The paper is generally well‐organized, and the findings are of practical relevance for tailoring HBV vaccination in populations that may be prone to lower immune responses. The manuscript is largely clear, and the conclusions are substantiated by the data.
Below are some specific comments and suggestions that could further strengthen the article:
You note that blood was collected only once (eight weeks) after the third dose. While this is a commonly used time point to assess peak antibody responses, please briefly justify in the Methods (or Discussion) why additional timepoints (e.g., one month after each dose or a long‐term follow‐up) were not collected. These would have provided valuable information on both early kinetics and durability of the response.
The paper identifies smoking and alcohol as influential factors for lower GMC in the YDV group but less so in CHO. It would be helpful to discuss possible mechanistic underpinnings a bit more (briefly, in the Discussion), and to highlight that the frequency or intensity of these habits was not captured. This limitation might be acknowledged more explicitly since “heavy,” “moderate,” or “light” usage could alter immune responses differently.
You used WHO cutoffs for overweight and obesity, which is standard. However, you merged all “underweight” and “normal” participants into a single category (≤ 24.9 kg/m^2). If possible, please clarify how many participants were actually underweight (BMI < 18.5) versus normal (18.5–24.9). If the underweight group is extremely small (which is often the case), a brief note confirming that no separate analysis was performed due to low numbers would be useful.
Although you excluded participants with major comorbidities, please highlight in your Discussion that other low‐level comorbidities or lifestyle factors (e.g., chronic conditions that do not exclude participation, dietary variations) could still influence response. Including an acknowledgment of these potential confounders would strengthen the Discussion.
The current study design does not address how rapidly titers might wane in each vaccine group. While you do mention the importance of higher initial GMC for long‐term protection, consider adding a sentence or two about the need for future longitudinal studies (e.g., 2–5 years post‐vaccination) to confirm which regimen confers more durable immunity.
Reviewer 3 Report
Comments and Suggestions for Authors
General comment
The paper has a very convincing concept. It correlates the strength of the vaccine response with the factors in the recipients which may it impair and at the same time they provide data with a more effective vaccine than the standard vaccine (which is used worldwide even for so demanding goals as prevention of MTCT). Here, middle-aged adults are targeted. The text describes in L246 the goal, citation from the discussion: “… the goal was to explore personalized vaccination strategies to enhance efficacy and prolong protection.” This goal was very well achieved.
However, the paper may gain much relevance at a worldwide level beyond China, if the aspects mentioned below are considered.
Specific points
- At least as important as the well-known relationship between age, body weight etc. and antiHBs response is the direct and very careful comparison of the two vaccine types. I suggest a modified title: Comparison of yeast and CHO cell-derived Hepatitis B Vaccines and Influencing Factors in Vaccine-Naïve Adults Aged in China: Insights for Personalized Immunization Strategies (23 words, originally 24)
- Comparison of two vaccines implies efficacy assessment.
- The exact age range is not important in the title. Adult is sufficient.
- Replace elevated by higher.
- L46 and 98-103. The description of the two vaccines in the text and in ref. 4 is not sufficient for an international readership with the main interest in vaccines.
- Which subgenotype had the HBV DNA used for the generation of the expression vectors? Were they identical for both cell types? What is the HBsAg subtype formula for the HBsAg, e.g. adw2 or adr?
- What was the % proportion of the three HBs protein in the CHO vaccine: LHBs with pres1, MHBs with preS2 and SHBs? At least an approximate number would necessary. An SDS gel of the CHO vaccine would be useful if no other data are available. Ref. 4 gives only the number of “monomers” per particles.
- What does that mean: “… with each group receiving alternate doses.”?
- L120-124. The Abbott assay for antiHBs detects down to 1 mIU/mL. Were the results 1-10 mIU/mL considered negative? Were they included in the calculation of the GMCs? Unfortunately, it is international consensus to consider <10 mIU/mL negative. But, if the authors have values 1-10, they should include them because they are also neutralizing antibodies though in low amounts. This low amount may also indicate that a booster would be possible.
- L144-152. The authors reported that the Abbott assay determines antiHBs levels 1- 1000 mIU/mL. But many GMCs exceed 1000. Were the sera diluted for antiHBs quantitation? (See also above). If not, an underestimation of the concentration may result.
- L202-207, L221-224 and 237-240. The seemingly smaller effect of BMI (or obesity) or smoking or alcoholism with the CHO vaccine may be caused by underestimation of high antiHBs concentrations. See above.
- Somewhere in the paper, brief definitions of overweight, obesity, smoking and drinking or alcoholism should be given, either in the study population or as footnotes to table 2 or fig. 2.
- L264 and L339. Concerning the variable outcome with YDVs in various studies, the authors write: the types of vaccines used may also contribute to the differing outcomes. They should mention that they used a YDV from Hansenula while most studies may have used a YDV from Saccharomyces. What is the potential advantage of Hansenula?
- The authors cite ref. 12 (Stramer et al, NEJM 2011) correctly that low antiHBs titer do not reliably avert HBV infection. But ref. 12 mentions also that the breakthrough HBV infections were caused by HBV genotypes different from the vaccine strain. They should mention this fact at an appropriate site and describe the HBV genotypes in the vaccines and in the population in China.
- One of the strengths of the CHO vaccine is that the preS domains contain several Th cell epitopes whereas the small HBs protein is a weak Th antigen.
- L332-334. What is the justification for a weaker vaccine? Lower price? What is the price difference in China?
